# Combined PARP and Dual Topoisomerase Inhibition Potentiates Genome Instability and Cell Death in Ovarian Cancer

**DOI:** 10.3390/ijms231810503

**Published:** 2022-09-10

**Authors:** Inken Flörkemeier, Julia S. Hillmann, Jörg P. Weimer, Jonas Hildebrandt, Nina Hedemann, Christoph Rogmans, Astrid Dempfle, Norbert Arnold, Bernd Clement, Dirk O. Bauerschlag

**Affiliations:** 1Department of Gynaecology and Obstetrics, Kiel University and University Medical Center Schleswig-Holstein Campus Kiel, 24105 Kiel, Germany; 2Pharmaceutical Institute, Department of Pharmaceutical and Medicinal Chemistry, Christian-Albrecht University of Kiel, 24118 Kiel, Germany; 3Institute of Medical Informatics and Statistics, Kiel University and University Medical Center Schleswig-Holstein Campus Kiel, 24105 Kiel, Germany

**Keywords:** ovarian cancer, dual topoisomerase inhibitor, PARP inhibitor

## Abstract

Although ovarian cancer is a rare disease, it constitutes the fifth leading cause of cancer death among women. It is of major importance to develop new therapeutic strategies to improve survival. Combining P8-D6, a novel dual topoisomerase inhibitor with exceptional anti-tumoral properties in ovarian cancer and compounds in preclinical research, and olaparib, a PARP inhibitor targeting DNA damage repair, is a promising approach. P8-D6 induces DNA damage that can be repaired by base excision repair or homologous recombination in which PARP plays a major role. This study analyzed benefits of combining P8-D6 and olaparib treatment in 2D and 3D cultures with ovarian cancer cells. Measurement of viability, cytotoxicity and caspase activity were used to assess therapy efficacy and to calculate the combination index (CI). Further DNA damage was quantified using the biomarkers RAD51 and γH2A.X. The combinational treatment led to an increased caspase activity and reduced viability. CI values partially show synergisms in combinations at 100 nM and 500 nM P8-D6. More DNA damage accumulated, and spheroids lost their membrane integrity due to the combinational treatment. While maintaining the same therapy efficacy as single-drug therapy, doses of P8-D6 and olaparib can be reduced in combinational treatments. Synergisms can be seen in some tested combinations. In summary, the combination therapy indicates benefits and acts synergistic at 100 nM and 500 nM P8-D6.

## 1. Introduction

Currently, ovarian cancer (OC) constitutes the fifth leading cause of cancer-related death among women in the developed world due to high therapeutic resistance, prolonged latency period until diagnosis and a lack of effective treatments [1,2,3,4]. Frequent diagnoses in advanced stages lead to poor prognoses. Despite advances in therapy, including therapy targeting the homologous recombination (HR) pathway, survival for patients with advanced disease is poor.

First line therapy for OC includes cytoreductive surgery, platinum- and taxane-containing chemotherapy with maintenance therapy of bevacizumab and olaparib or niraparib and symptom-oriented clinical follow-up care [5]. In addition, patients benefit from second line therapy with topoisomerase inhibitors such as liposomal doxorubicin or in further lines with topotecan. However, currently the response is not sufficient [6,7]. Consequently, it is a necessity and an important aim to reduce mortality by improved new therapeutic options.

A new promising compound for OC therapy is the dual topoisomerase inhibitor P8-D6. P8-D6 was investigated as a single-drug in in vitro studies in a translational approach and showed high antitumoral efficacy [8,9]. P8-D6 is an inductor of apoptosis by acting as a dual topoisomerase poison. It covalently stabilizes the enzyme-cleaved DNA complex of both topoisomerases (topoisomerase I/II) [10]. Topoisomerase regulates torsional stress in DNA by inducing single- and double-strand breaks (SSB/DSB), which is important for DNA unwinding to enable essential genome functions (e.g., transcription, replication or recombination) [11,12,13]. Thus, topoisomerases play a major role in DNA replication and chromosome condensation. Stabilization of topoisomerase I/II-DNA complexes causes an increased set of DNA strand breaks and leads to more unstable DNA replication [14]. Topoisomerase poisoning leads to cell death by inducing apoptosis. As topoisomerase expression is deregulated in OC [15,16], it is an interesting target to inhibit. P8-D6 is an aza-analogous Benzo[*c*]phenanthridine with cytotoxic properties [17]. Its high efficacy in the therapy of OC and breast cancer was recently demonstrated in 2D monolayer cell culture and spheroids [8,9].

Since 2019, PARP (poly-ADP-ribose polymerase) inhibitors are approved for therapy of recurrent OC after response to platinum reinduction, regardless of BRCA pathogenic variant status [18]. These prolong the progression-free interval, in several cases considerably, and serve as maintenance therapy [19]. PARP enzymes are involved in repairing damaged DNA (Figure 1). DNA repair mechanisms play a key role in maintaining genomic integrity and, consequently, cell survival. DNA SSB can be induced directly by reactive oxidative species or can occur as a consequence of topoisomerase I-DNA intermediates [20,21]. PARP builds a complex with tyrosyl-DNA phosphodiesterase 1 (TDP1), which stimulates the excision of the covalent topoisomerase I-DNA complexes. Subsequently SSBs can be repaired [22]. The repair of SSBs proceeds by base excision repair (BER) or nucleotide excision repair (NER), mechanisms in which PARP enzymes play a major role [23,24]. In BER DNA, strand breaks are identified by PARP. Further repair enzymes are recruited and, subsequently, the ssDNA gap is closed by XRCC1, DNA polymerase β and ligase 3. DSBs arise spontaneously during physiological DNA processing or are caused by agents or radiation. The two major pathways of DSB repair are HR and nonhomologous end joining (NHEJ) [23,24]. In HR repair, PARP promotes the recruitment of MRN complex to collapsed replication forks and the accessibility of repair factors to damage by modifying chromatin, as well as recruiting and activating ATM kinase [25,26]. ATM kinase phosphorylates checkpoint and repair pathway proteins such as H2AX histone protein. The phospho-H2A.X (γH2A.X) forms foci with other protein complexes in the region of DSBs. This in turn triggers the activation of nucleases such as Mre11 and CtIP to process the DNA ends and to generate ssDNA with 3′ overhangs. In the next step, the recombinase RAD51 match for a homologous sequence in the sister chromatid. This serves as a template for an error-free resynthesis of the damaged DNA [27]. γH2A.X and RAD51 are used as biomarkers for DNA strand breaks and HR repair. As PARP plays a key role in many repair pathways, many parts are disrupted by PARP inhibitors (PARPi).

The induction of strand breaks by topoisomerase poisons and the simultaneous inhibition of DNA repair by olaparib represents a plausible approach for an extended benefit in cancer treatment response. For example, PARP-1 promotes the cleavage of topoisomerase I from DNA and thus DNA repair. P8-D6 induces SSBs by topoisomerase I inhibition, which convert to DSBs in the absence of functional PARP. Additionally, the inhibition of Topo II leads to DSB. DSB can be repaired by HR. If there is an HR deficiency, cells induce increased apoptosis. Previous studies demonstrated for topoisomerase I inhibitors that inactivation of PARP leads to sensitization to topoisomerase I toxins in cancer cells [28,29,30,31]. However, initial studies with a selective topoisomerase II inhibitor also showed an advantage in combination therapy with PARPi [32]. If the HR system is disturbed simultaneously, the effect of PARPi will be increased. Thus, the determination of the HR deficiency score could predict therapy response and provide an additional benefit for therapy. The HR deficiency score integrates the DNA-based measures of genomic instability in the form of loss-of-heterozygosity, telomeric allelic imbalance and large-scale state transitions. The parameters are selected as follows: (1) number of LOH larger than 15 Mb but shorter than whole chromosome; (2) number of telomeric allelic imbalances (TAI) at the very end of the chromosome arm, larger than 11MB and not crossing the center; (3) number of breakpoints in large scale transitions (LST) larger than 10 MB, ignoring regions shorter than 3 MB. HRD scores ≥42 countable events in a genome were considered to be HR deficient. In addition to increased efficacy, less development of resistances can be observed in combinational treatments, since two different mechanisms of action are addressed.

Our study investigated whether the highly effective dual topoisomerase inhibitor P8-D6 in combination with a PARPi olaparib can provide an advantage for the treatment of OC. This is the first study to suggest the novel dual topoisomerase inhibitor P8-D6 as an enhancer drug for extending the therapeutic spectrum of PARPi.

## 2. Results

To evaluate whether dual topoisomerase inhibition increases the cytotoxicity of PARPis we treated OC cells with either increasing concentrations of the single agents olaparib, P8-D6 or with the combination of both. P8-D6 induces SSBs and DSBs and olaparib prevents strand-break repair. In our investigated cell lines, we found a high HR deficiency (HRD score: 70) in OvCar8, and for A2780 cells a low HRD score of 5, respectively. Gene sequencing revealed a pathological alteration in the TP53 gene (c. 376-1G > A) for OvCar8 cells and a pathogenic variant in the PIK3CA gene (c. 1093G > A) for A2780 cells. No BRCA pathogenic variance was detectable.

### 2.1. P8-D6 Sensitizes Ovarian Cancer Cells to Olaparib in 2D Cell Culture

To identify the best schedule for combinatorial treatments we ran several validation pretests with different treatment designs. The best setting was a sequential treatment with olaparib, added 24 h after seeding and P8-D6 supplemented 24 h later. Thus, the total incubation time for olaparib was 72 h and for P8-D6 it was 48 h (Figure 2A). After treatment with one or both inhibitors, cell viability and caspase activity were assessed. P8-D6 significantly increased olaparib toxicity in OvCar8 and A2780 cells in 2D monolayer. Using OvCar8 cells, the addition of 10 µM olaparib reduced the IC_50_ value of P8-D6 from 151.9 nM to 32.82 nM (Figure 2B). In A2780, the change in P8-D6 IC_50_ value was between 55.9 nM without olaparib and 0.33 nM with 10 µM olaparib (Figure 2D). Considering the benefit of P8-D6 for olaparib therapy, the addition of 500 nM P8-D6 reduced the olaparib IC_50_ value from 11.1 µM to 0.03 µM in OvCar8 cells (Appendix A). For A2780 cells, the olaparib IC_50_ value was reduced from 1.42 µM to 0.003 µM (Appendix A). In addition, the combinatorial treatment resulted in significantly higher caspase activity levels than single-drug therapy (Figure 2C,E). Interestingly, despite the low HR deficiency, A2780 cells appeared to be very sensitive to olaparib and P8-D6. Taken together, these results suggest that olaparib inhibits cell proliferation and that P8-D6 enhances this apoptotic effect.

Further, the combination index (CI) was calculated to assess benefits of combinational treatment in OvCar8 cells and A2780 in 2D monolayer. Synergistic effects of the combination treatment with olaparib und P8-D6 on cell viability was observed with 500 nM P8-D6 in OvCar8 cells and with 100 nM P8-D6 in A2780 cells (Figure 3A, Appendix A). The calculated DRI values demonstrate that most cells were more sensitive to olaparib when combined with P8-D6, compared with single-drug therapy (Figure 3B). Thus, in the OvCar8 2D monolayer 0.5 µM olaparib can be reduced by 9.87-fold when treated in combination with 0.05µM P8-D6, reaching the same efficacy. Similar investigations and results can be seen for A2780 cells in Appendix A.

### 2.2. P8-D6 Sensitizes Ovarian Cancer Cells to Olaparib in 3D Cell Culture

The spheroid models depict physiologically relevant biological structures more realistically than a 2D monolayer cell culture does. These spheroid systems are easy to use, modifiable and reproducible, which makes them the preferred model for drug screens. In order to determine whether the benefit of combination therapy can be seen in 3D-spheroids as well, we again compared single-drug therapy with combination therapy. To assess olaparib and P8-D6 combination effects, viability and caspase in OvCar8 spheroids were determined using previously established methods [8,33,34,35]. Interestingly, a higher concentration of both drugs was needed in the 3D spheroids to achieve the same effect as the 2D. However, in a slightly higher concentration range, a significant additional benefit is also demonstrated by P8-D6 in the spheroids. The IC_50_ value of P8-D6 in combination with 10 µM olaparib was 3,5-fold lower compared with single-drug treatment in the spheroids (Figure 4B). In addition, significantly higher caspase activities were detected in some combinations compared with P8-D6 only (Figure 4C). A markable increase in efficacy could also be detected for the addition of P8-D6 to olaparib treatment (Appendix A). The results were interpreted by CompuSyn software to assess the CI in the OvCar8 3D-spheroids, reflecting synergistic outcomes (Figure 5A). Moreover, this calculation determined that the dose of olaparib can be reduced in the combination therapy, especially in combination with 0.1 and 0.5 µM P8-D6 and at the established concentrations ≥5.0 µM olaparib (Figure 5B). 

Further on, the cytotoxicity of the spheroids after incubation with olaparib and P8-D6 was investigated using CellTox™ Green dye. Results suggest a decreased viability of spheroid cells after treatment with the combination compared with single-drug treatment (Figure 6A,B). The fluorescence intensity increases over time, suggesting increased cell death induced by olaparib and P8-D6. Further studies of cytotoxicity involve staining with Calcein AM/propidium iodide as a marker for viable/dead cells. The combination of 10 µM olaparib with 0.1 µM P8-D6 was notably more effective in target cell cytotoxicity (Figure 6C). Scanning electron microscopy analyses of the treated spheroids show strong changes in membrane integrity in combination therapy compared with single-drug treatment (Figure 6D).

### 2.3. Combined Treatment of Olaparib and P8-D6 Leads to Genome Instability

To observe responses to DNA damage after treatment, γH2A.X and RAD51 expression was evaluated by immunofluorescence (Figure 7). Inductions of γH2A.X and RAD51 were considered as biomarkers for DNA strand breaks and HR repair. OvCar8 and A2780 cells were treated in a 2D monolayer with olaparib (2.5 µM, 10 µM), P8-D6 (0.1 µM) and a combination of both. Subsequently, they were stained with γH2A.X and RAD51 antibodies. The intensities of these DNA-damage markers were remarkably higher in combined therapy compared with monotherapy (selected concentrations Figure 7A,B (OvCar8); all concentrations Appendix A (OvCar8); Appendix A (A2780)). As 3D cell cultures more precisely mimic the natural cell microenvironment and diffusion of drugs, staining with γH2A.X and RAD51 was also performed with the spheroids (Figure 7C,D and Appendix A). To validate the method, we tried different ways of staining the spheroids. Best results were achieved with glycerol, which made the spheroids more permeable for signals (Appendix A). Analogously to the results of the 2D immunofluorescence analyses, the fluorescent signal for DNA damage in spheroids treated with olaparib and P8-D6 was remarkable higher than with 2.5 µM olaparib only (Figure 7C,D). The higher concentration of 10 µM olaparib in combination therapy generated a lower γH2A.X and RAD51 signal than the combination with 2.5 µM olaparib due to increased instability of the spheroids (Appendix A). This demonstrates that the DNA damage caused by P8-D6 cannot be fully repaired by PARP.

### 2.4. Combination Therapy Changes Protein Expression of Marker Molecules

To further understand and prove the mechanisms of action of combination therapy, the protein expression of several marker molecules was assessed by Western blots (Figure 8). OvCar8 cells were treated in 2D monolayers with olaparib (2.5 µM, 10 µM), P8-D6 (0.1 µM) and a combination of both. Protein expression was determined in cell lysates. According to the results of the immunofluorescence intensities of the treated 2D OvCar8 cells mentioned before, similar changes in the expression of γH2A.X and RAD51 were seen in Western blot. The expression of RAD51 was the highest in OvCar8 treated with 0.1 µM P8-D6 and the lowest, apart from the PBS control, in 2.5 µM olaparib. In the combination therapy, the expression of RAD51 was higher than in the monotherapy with olaparib, but still lower in expression than the monotherapy with P8-D6. An additional benefit of the combination therapy was seen in γH2A.X expression, too. The additional treatment with P8-D6 led to a clear increase of γH2A.X expression compared with the monotherapy with olaparib. In addition, the expression of topoisomerase II α, topoisomerase I, PARP1 and its cleaved form were investigated, which are all targets of the used cytostatics. Upregulation of topoisomerase II α expression was determined in samples treated with the dual topoisomerase inhibitor. Since the expression of topoisomerase I was already at a high level, we were not able to detect an increase. OC cells in total expressed a high level of topoisomerase I [8]. Contrary to this, changes in topoisomerase II α expression became evident, caused by treatment, whereas no obvious differences could be seen in PARP1 expression levels. Cleaved PARP1, which is a fission product of PARP1 that occurs during the process of apoptosis due to activity of caspase3, represented changes in about the same range as for apoptosis. Interestingly, the combination with 2.5 µM olaparib caused an increase in cleaved PARP1 but, with 10 µM olaparib treatment, no gain was detectable. Surprisingly, the activity of caspase seemed to be the highest in the cells treated with 10 µM olaparib. Another interesting molecule to investigate the efficacy of therapy is p53, a tumor suppressor protein that can promote apoptosis and cell cycle arrest. Cells treated with the combination therapy showed higher signal intensities compared with the monotherapy with olaparib. To further optimize therapy strategies, a plausible approach could have been to additionally inhibit TDP1 or TDP2, as TDP interferes with the repair of topoisomerase DNA complexes. To analyze whether an additional inhibition made sense or not, Western blots were done. These clearly showed that TDP1, as well as TDP2, signal intensities decreased with combinational treatments. Thus, there is less TDP to inhibit by TDP1 inhibitors. In summary, P8-D6 sensitizes ovarian cancer cells to olaparib in 2D and 3D cell cultures. Combined therapy leads to genome instability and to protein expression changes of marker molecules.

## 3. Discussion

The purpose of the current study was to look for a synergistic effect of olaparib in combination with P8-D6 in OC. P8-D6 is a novel dual topoisomerase poison which stabilizes the topoisomerase DNA-complex and induces more and permanent SSB and DSB [10]. In single-drug therapy, P8-D6 already showed an efficacy for the treatment of OC and breast cancer [8,9]. In this present study, we presented that P8-D6 causes a beneficial interaction with olaparib regardless of the HR deficiency status and increases genetic instability.

By developing PARPis, compounds were designed and approved that act by synthetic lethality. OC is characterized by genetic abnormality, including alterations of DNA damage repair pathways, particularly in BER and HR deficiency. These are targets for PARPi. Its efficacy has already been verified in clinical studies. SOLO-1 demonstrated that the progression-free survival (PFS) of women with newly diagnosed OC with pathogenic variant in BRCA was improved when treated with olaparib [19]. The PRIMA trial provided data on the benefit of niraparib in patients with advanced OC regardless of BRCA pathogenic variant status in the first line setting [36]. The PARPi Veliparib was investigated mainly in combination with chemotherapy, but has not received approval for clinical use (VELIA) [37,38]. Even though PARPi therapy is currently used as maintenance therapy in first line therapy, side effects and the development of resistances are possible. Side effects such as myelosuppression have previously been described for olaparib and niraparib [39,40].

Cancer drugs are most effective when given in combination [41]. The basic idea of combination therapy is to use drugs with different mechanisms of actions, thereby decreasing development of resistances and side effects. Novel treatment strategies for advanced OC combine PARPis and other anticancer therapies based on DNA repair mechanisms. Several compounds have already been extensively studied in combination with PARPis, including immune checkpoint inhibitors (durvalumab [42]), antiangiogenic agents (bevacizumab, cediranib [43,44]), PI3K inhibitors (buparlisib), AKT inhibitors (capivasertib) and chemotherapy (paclitaxel, carboplatin, topotecan [45,46,47]). In the clinical study, PAOLA-1 Bevacizumab improved PFS when combined with olaparib as a maintenance therapy [44].

To further increase the potential of olaparib on OC cells, a study was performed in combination with P8-D6. We reported that the addition of olaparib showed a significant beneficial caspase induction at low nanomolar concentrations for P8-D6 treatment in OvCar8 and A2780 cells. It was also found that A2780 cells were more sensitive to P8-D6 compared with OvCar8 cells. Gao et al., [48] determined the additional benefit of olaparib with cisplatin and paclitaxel in A2780 and OvCar3 cells. After 48 h incubation, the olaparib monotherapy triggered an IC_50_ value of 6 µM in A2780 cells for olaparib. Compared with this, our assay achieved an IC_50_ value of 1.142 µM for olaparib after 72 h. These variations can be due to the time of treatment. The combination of PARPi with chemotherapeutics is based on the DNA-damaging effect of the chemotherapeutic drug with a simultaneous disruption of DNA-repair by the PARPi. This leads to increased DNA-damage and an enhanced anti-tumor effect. According to this, an increased effectiveness makes dose reduction possible, as we proved in this study in the 2D monolayer and the spheroids. Improvement was reached in almost all samples by the addition of olaparib. The evaluation of cytotoxicity and membrane integrity provided additional important insight, as the 3D structure included effects of cellular architecture and drug penetration. Our results also verified clear effects and changes by combination therapy in 3D.

Olaparib was combined with chemotherapy to interfere BER and HR by PARP inhibition. As mentioned, olaparib has already been investigated in combination with carboplatin and paclitaxel [45], pegylated liposomal doxorubicin [49] and topotecan [46] in clinical trials. Gao et al., [48] investigated the effectiveness of the cisplatin with olaparib in OC. Results showed that a combination treatment with olaparib and cisplatin or olaparib and paclitaxel significantly inhibited tumor growth compared with a control and a single treatment. This indicates that the induction of DNA damage and DNA repair is closely connected and influences the mutual effectiveness. Since topoisomerase inhibition is responsible for catalyzing the formation of DNA strand breaks, it has already been investigated whether the combination of topo inhibitors with PARPis have an additional benefit [22,24,29,50,51]. To confirm, the combination study with ABT-888 and irinotecan was performed, indicating a synergistic effect in the treatment of colon cancer cells with PARPis and topoisomerase I inhibitors [52]. Moreover, the use of olaparib and SN-38, an active metabolite of irinotecan, showed a synergistic effect in colon cancer cells and an enhancement of the RAD51-mediated HR signaling pathway [53]. Eetezadi et al., [54] investigated the synergism of doxorubicin and olaparib treatment in 2D and 3D-spheroid OC models. They presented that in, certain combination ratios, doxorubicin–olaparib combinations were highly effective in inhibiting OC cell growth. The ratios 1:2 and 1:5 (doxorubicin: olaparib) showed antagonistic effects; in contrast 1:10, 1:100 were synergistic. For topoisomerase II inhibitors and PARPi combinations, the results are more oppositional. Bowman et al., measured in leukemia cells no potentiation of etoposide cytotoxicity by PARPi [31]. However, further studies could show for the selective topoisomerase II inhibitor C-1305 that PARP inhibition potentiates cytotoxicity [32]. Interestingly, Yuan et al., developed a single drug that has inhibitory activities against topoisomerase and PARP-1 and was tested on breast cancer cells [55]. Treatment studies on OC cells using a dual topoisomerase inhibitor and a PARPi have not been published so far.

Furthermore, we visualized DNA damage in the 2D monolayer and the spheroids accumulating after P8-D6 and olaparib treatment by detecting RAD51 and γH2A.X, two DNA damage markers. The determination of γH2A.X intensity is a reliable biomarker for DSBs. The accumulation of RAD51 is described as a functional biomarker for HR deficiency. RAD51 and γH2A.X accumulation was increased by our combination therapy. Among SSB pathways BER is the primary method and PARP is essential for this process. In the absence of PARP, SSBs will ultimately be converted to DSBs and be repaired by HR [23]. P8-D6 induces SSBs by topoisomerase I inhibition, which convert to DSBs in absence of functional PARP (Figure 1). This explains the increase in RAD51 and γH2A.X accumulation. However, this effect was not clearly detectable in the spheroids with 10 µM olaparib and 0.1 µM P8-D6 combination, as the spheroids visibly lost integrity. The expression analyses detected an upregulation of all target proteins: topoisomerase I, topoisomerase II and PARP in the initial state or with treatment. Enzymes representing efficacy were upregulated in the combination therapy. Only TDP 1 and TDP 2 were downregulated, therefore making them less interesting as further targets for inhibition.

This is the first study investigating dual topoisomerase inhibitors and PARPi in OC. The combination of PARP with P8-D6 is an interesting and promising therapeutic strategy. Inhibition with P8-D6 leads to SSBs and DSBs, which can only be mended by functional DNA repair pathways (Figure 1). Thus, these have a great influence, and the inhibition of DNA repair by PARP and the impact of pathogenic variants can affect the success of therapy. Both the already approved drug olaparib and the novel drug P8-D6 in combination indicate sensitization of cells in therapy and obtain great benefit for OC therapy.

## 4. Materials and Methods

P8-D6 was synthesized as recently described [17] and solved in PBS. Olaparib (AZD 2281, Axon Medchem) was solved in PBS with 0.05% DMSO.

### 4.1. Cell Preparation and Culture

The human OC cell lines OvCar8 and A2780, purchased from American Type Culture Collection (ATCC), were maintained in RPMI 1640 medium, including L-glutamine supplemented with 10% FBS, 60 IU(µg)/mL penicillin–streptomycin. Cells were grown at 37 °C and 5% CO_2_ in a humidified incubator, and subcultivated by a confluency of 70–80%. Cell line authenticity was checked by STR-profiling as described previously [56], and mycoplasma contamination was routinely investigated, using MycoAlert™ (Lonza).

### 4.2. 2D viability and Apoptosis

For the measurements of viability and apoptosis, cells of approximately 2000/well were seeded in a 96-well plate (Corning 3903). Cells were treated with olaparib on the following day. After a further 24 h, treatment was supplemented with P8-D6. Thus, the total incubation time for olaparib was 72 h and for P8-D6 it was 48 h. This treatment scheme was previously optimized in preliminary tests. The measurement was performed using ApoLive-Glo™ Multiplex Assay (Promega G6410) as described in the instruction (TM325). Viability was measured in fluorescence units (400_Ex_/505_Em_), and Caspase-Glo 3/7 cleavage was determined as relative luminescence units (RLU) using a microplate multimode reader (Spark, Tecan). For relative caspase activity calculation, caspase activity was divided by the viability (normalized to control). With viability data, dose–response curves were plotted using four parameter logistic regressions and inhibitory concentration 50% (IC_50_) values were calculated (GraphPad Software, Inc., San Diego, CA, USA).

### 4.3. 3D Viability, Apoptosis and Cytotoxicity

A2780 (200/well) and OvCar8 (2000/well) cells were seeded into a 96-well ultra-low attachment plate (Corning 4520) and maintained for 72 h. Then, half of the medium was removed and the spheroids were treated with olaparib. After a further 24 h, P8-D6 treatment was added. Thus, the total incubation time for olaparib was 72 h and for P8-D6 it was 48 h. Simultaneously to the P8-D6 treatment, CellTox™ Green assay (Promega G8731) was added. Cytotoxicity was detected (485_Ex_/520_Em_) and quantified 24 h and 48 h after addition, using NYONE^®^ Scientific (SYNENTEC) with 4× magnification. The following excitation sources and emission filters were used: Brightfield BF_Ex_/Green_Em_ (530/43 nm); CellToxTM Green (Blue_Ex_ (475/28 nm)/Green_Em_ (530/43 nm)). Following the treatment, the viability and the apoptosis of the cells were determined by RealTime-Glo™ (460_Em_) (Promega G9711) and Caspase-Glo 3/7 (565_Em_) (Promega G8090), using a microplate multimode reader (Spark, Tecan). The measurement was performed according to the instructions. The viability values were used to normalize the caspase results (relative caspase activity: caspase activity divided by the viability (normalized to control). With viability data, dose–response curves were plotted and inhibitory concentration 50% (IC_50_) values were calculated (GraphPad). For live–dead staining, cells were grown and treated as described above. Then, part of the medium was removed and replaced with propidium iodide (PI) (10 µg/mL), calcein-AM (1 mM) and Hoechst 33342 (0.001%) in medium for 3 h. For imaging, the NYONE^®^ Scientific (SYNENTEC) microscope was used, with 4x magnification and with the following excitation sources and emission filters: Brightfield BF_Ex_/Green_Em_ (530/43 nm); Hoechst 33342 UV_Ex_ (377/50 nm)/Blue_Em_ (452/45 nm); calcein-AM Blue_Ex_ (475/28 nm)/Green_Em_ (530/43 nm); propidium iodide Lime_Ex_ (562/40 nm)/Red_Em_ (628/32 nm).

### 4.4. Combination Index and Dose-Reduction Index

To identify synergistic drug combinations, data were analyzed using CompuSyn (ComboSyn Inc.; Cambridge, UK). Combination index (CI) and dose-reduction index (DRI) values were determined for each drug combination and single-drug therapy. Therefore, viability was normalized to the highest viability and subsequently converted to fraction affected values and analyzed with CompuSyn software [57,58]. CI values indicate the effect of combining multiple drugs (synergistic, additive or antagonistic); DRI values represent the fold decrease in the dose of a drug needed in a combination to achieve the same efficacy (fa) as the drug alone. Using this approach, drug combinations with CI values  <  1 are synergistic, CI values = 1 are additive, CI values > 1 are antagonistic and DRI values  >  1 are favorable.

### 4.5. Scanning Electron Microscopy

The spheroids were grown and treated as described in section 3D viability, apoptosis and cytotoxicity. Then, the spheroids were fixed with 2.5% glutaraldehyde for 1 h at RT. After washing, the second fixation was performed using 1% osmium tetroxide for 1.5 h at RT. The spheroids were washed and dehydrated with ethanol [25%, 50%, 75%, 96%, 100%] and air dried using hexamethyldisilazane on charcoal stubs overnight. For better conductivity, the spheroids were coated with gold and subsequently measured with scanning electron microscopy (Phenom XL, Phenom-world).

### 4.6. Fluorescence Imaging (LSM)

For the 2D monolayer culture, A2780 (15,000/well) and OvCar8 (10,000/well) cells were seeded in glass-bottomed 8-well chamber slides (Ibidi 80807). Cells were treated with olaparib 24 h after seeding. After a further 24 h, treatment was supplemented with P8-D6. Thus, the total incubation time for olaparib was 72 h and for P8-D6 it was 48 h. After treatment, the cells were washed with PBS, fixed with 4% paraformaldehyde (10 min at RT) and permeabilization using 0.2% Triton X-100 (5 min at RT). After blocking with goat serum (1:20 in PBS, 1 h at RT) cells were incubated with primary antibodies (anti-RAD51 1:100 (Proteintech 67024-1-Ig), anti-γH2A.X 1:100 (cell signaling 9718)) at RT for 1.5 h. Secondary antibodies (Goat anti-Mouse IgG (H + L) (Alexa Fluor 594 1:1000 (ThermoFisher A-11032), Goat Anti-Rabbit IgG (H + L) (Alexa Fluor^®^ 488) 1:1000 (abcam ab150081)) were stained for 1.5 h at RT. Cells were washed with PBS and DAPI/mounting medium (0.5 µg/mL) (Vectashield) was added.

For the 3D culture, OvCar8 (3000/well) cells were seeded into a 96-well ultra-low attachment plate (Corning 4520) and maintained for 72 h. Then half of the medium was removed and the spheroids were treated with olaparib. After another 24 h, P8-D6 was added. Thus, the total incubation time for olaparib was 72 h and for P8-D6 it was 48 h. The spheroids were fixed and incubated with an antibody as described previously [59]. The spheroids were transferred into PCR-tubes, washed and fixed with 4% PFA (1 h at 37 °C), followed by washing with PBS supplemented with 1% FBS. Then, the spheroids were incubated with 0.5 M glycine in PBS for 1 h at 37 °C and with a penetration buffer (0.2% Triton X-100, 0.3 M glycine and 20% DMSO in PBS) for 30 min RT, to improve the penetration of antibodies and nuclear dyes. After this, the spheroids were washed and incubated in a blocking buffer (0.2% Triton X-100, 1% BSA, 10% DMSO in PBS) for 2 h at 37 °C. Primary antibodies (anti-RAD51 1:50 (Proteintech 67024-1-Ig) and anti-γH2A.X 1:100 (cell signaling 9718)) were diluted in an antibody buffer (0.2% Tween 20, 10 μg/mL heparin, 1% BSA, 5% DMSO in PBS) and incubated overnight at 37 °C. The spheroids were washed in a washing buffer (0.2% Tween 20, 10 μg/mL heparin, 1% BSA in PBS) and stained with a secondary antibody. Secondary antibodies (Goat anti-Mouse IgG (H + L) (Alexa Fluor 594 1:1000 (ThermoFisher A-11032), Goat Anti-Rabbit IgG (H + L) (Alexa Fluor^®^ 488) 1:1000 (abcam ab150081)) and Hoechst 33342 (1:5000) were diluted in the antibody buffer and incubated for 5 h at 37 °C. The spheroids were washed with the washing buffer and incubated in 88% glycerol overnight at 37 °C as a clearing procedure. All samples were gently shaken throughout all incubations. 

Fluorescence imaging was performed using the Zeiss LSM 880 microscope (Carl Zeiss Microscopy). For evaluation, the ZEN 2.5 software (blue edition) was used.

### 4.7. Western Blot Analysis

For the detection of protein expression, OvCar8 (60,000/well) cells were seeded on a 6-well plate (Cellstar 657160). Cells were treated with olaparib (2.5 µM, 10 µM) and P8-D6 (0.1 µM), as described above. After treatment, the cells were mechanically detached and washed with ice-cold PBS. Cell pellets were extracted in a lysis buffer (50 mM Tris pH 7.4, 150 mM NaCl, 2 mM EDTA, 1% Triton X-100, 1% NP-40, a protease inhibitor cocktail) supplemented with 1 mM sodium fluoride (NaF), 1 mM sodium orthovanadate (NaVO_3_) and phosphatase inhibitor cocktail by incubation at 4 °C for 20 min, followed by centrifugation at 12,000× *g*, 4 °C for 20 min. The soluble supernatant was then further analyzed. The amount of total protein was determined by detergent compatible protein assay (BioRad, 5000122). Samples were denatured with 4 X LDS buffer (Invitrogen™, NP0007) and 1 M Dithiothreitol and heat inactivated at 70 °C for 10 min. Equal amounts of protein (10 μg) were separated by electrophoresis on a 4–12% TrisGel (NuPAGE, Invitrogen™, NP0336BOX; PageRuler™ Thermo Scientific™ 26619) with MOPS buffer and transferred to PVDF-membranes (Millipore IPFL00010). Membranes were blocked in TBS supplemented with Tween-20 (TBST) and 5% (w/v) milk powder for 1 h. For analysis, membranes were incubated with primary antibodies at 4 °C overnight (anti-Topo I 1:500 (Santa Cruz sc-271285), anti-Topo II α 1:1000(cell signaling 12286), anti-PARP1 1:1000 (cell signaling 9542), anti-RAD51 1:2000 (Proteintech 67024-1-Ig), anti-γH2A.X 1:1000 (cell signaling 9718), anti-TDP1 1:1000 (Proteintech 10641-1-AP), anti-TDP2 1:500 (Proteintech 12203-1-AP), anti-p53 1:1000 (Santa Cruz sc-126), anti-HSP 90 1:5000 (Santa Cruz sc-13119), anti-β-Actin 1:1000 (Santa Cruz sc-47778) and then washed with TBST. Further, the membranes were incubated with HRP-labeled anti-mouse IgG 1:1000 (cell signaling 7076S) or HRP-labeled goat anti-rabbit IgG 1:1000 (cell signaling 7074S). After washing with TBST, membranes were developed with ECL substrate (Biorad 1705061) or SuperSignal^TM^ West Femto maximum sensitivity substrate (Thermo Scientific 34096). The protein levels of HSP 90 and β-Actin were used as loading controls.

### 4.8. HRD by aCGH

Array-based comparative genome hybridization (aCGH) was used to determine the degree of homologous repair deficiency (HRD). The aCGH was performed on 4 × 180 k CGH + SNP arrays from Agilent Technologies, whose SNP backbone was strengthened and which have a higher resolution for the genes BRCA1/2 and TP53 (design: 086822_V3_HRD_4x180k_CGH_SNP_Kiel1). Hybridization followed the standard protocol for “Array-Based CGH for Genomic DNA Analysis, version 7.4”. The fully hybridized and washed arrays were detected on a Sure Scan Microarray Dx scanner from Agilent and evaluated using the Agilent CytoGenomics analysis software (version 3.0.6.6). A total of 500–1000 ng of DNA from each of the cell lines OVCAR8 and A2780 were used. An equivalent amount of reference DNA (Agilent female) was used to compare. HRD status was determined by the definition published by Timms et al., in 2014 [60].

### 4.9. Gene Sequencing

Cells were harvested for isolation of DNA using QIAamp DNA Mini Kit (Qiagen, 56304, Hilden, Germany). DNA concentration was 10 ng/µL. Samples were sequenced by TruRisk^®^ Genpanel, as described previously [61].

### 4.10. Statistical Analysis

Statistical tests were performed using GraphPad Prism9 (GraphPad). Gaussian distribution was tested by the Shapiro–Wilk normality test. Data of multiple groups were checked with one-way ANOVA for statistical significance. Statistically significant differences were assumed at *p*-values < 0.05 (*) according to Dunnett multiple comparison.

## 5. Conclusions

In summary, this report shows the challenges of finding an effective treatment for OC and highlights new potential therapeutic strategies. The combination therapy of topoisomerase inhibitor and PARPi offers an advantage over PARP inhibition or topoisomerase inhibition alone in OC cells. This is the first study to propose the novel drug P8-D6 as an enhancer compound for extending the therapeutic spectrum of PARPi. Nevertheless, further studies are needed to elucidate the mechanisms behind the synergistic effect of olaparib combined with P8-D6 on OC cells in vitro and in vivo.

## 6. Patents

Clement, B.; Meier, C.; Heber, D.; Stenzel, L. novel pyrido [3,4-c] [1,9] phenanthroline and 11,12-dihydropyrido [3,4-c] [1,9] phenanthroline-derivatives and their use, in particular, for the treatment of cancer. PCT/EP2013/057212; granted for Europe, USA, Canada, Australia EP2834240, US9062054, CA 2869426, AU2013244918. Clement, B.; Meier, C.; Steinhauer, T.N. novel pyrido-phenanthroline derivatives, production and use thereof as medicaments. PCT/EP2017/080327; granted for Europe-EP3330270.

## Figures and Tables

**Figure 1 ijms-23-10503-f001:**
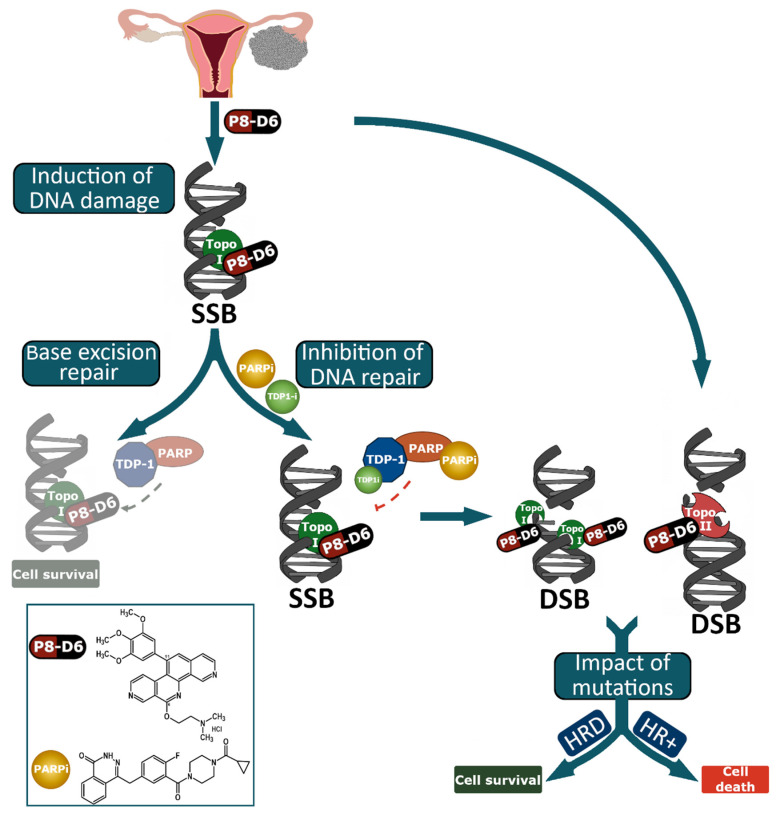
Mode of action of dual topoisomerase inhibition and PARP inhibition. The schematic representation shows the proposed targets of PARPi and topoisomerase inhibitors. PARP is involved in different DNA repair pathways, including BER and HR. Here, PARP contributes to the detection of lesions and the initialization of repair. Combining PARPi with P8-D6 causes an increase in genomic instability. P8-D6 stabilizes the topoisomerase I and II-DNA covalent complex, thereby leading to an increase of DNA damage in the form of SSBs and DSBs. By SSB repair, if DNA repair functions via TDP1 and PARP, the cell survives. Inhibition of PARP and/or TDP1 results in the accumulation of SSBs and DSBs, which can be repaired by HR. Moreover, the inhibition of Topo II generates DSB. If there is an HR deficiency, cells cannot be repaired via the HR pathway and go into induced apoptosis.

**Figure 2 ijms-23-10503-f002:**
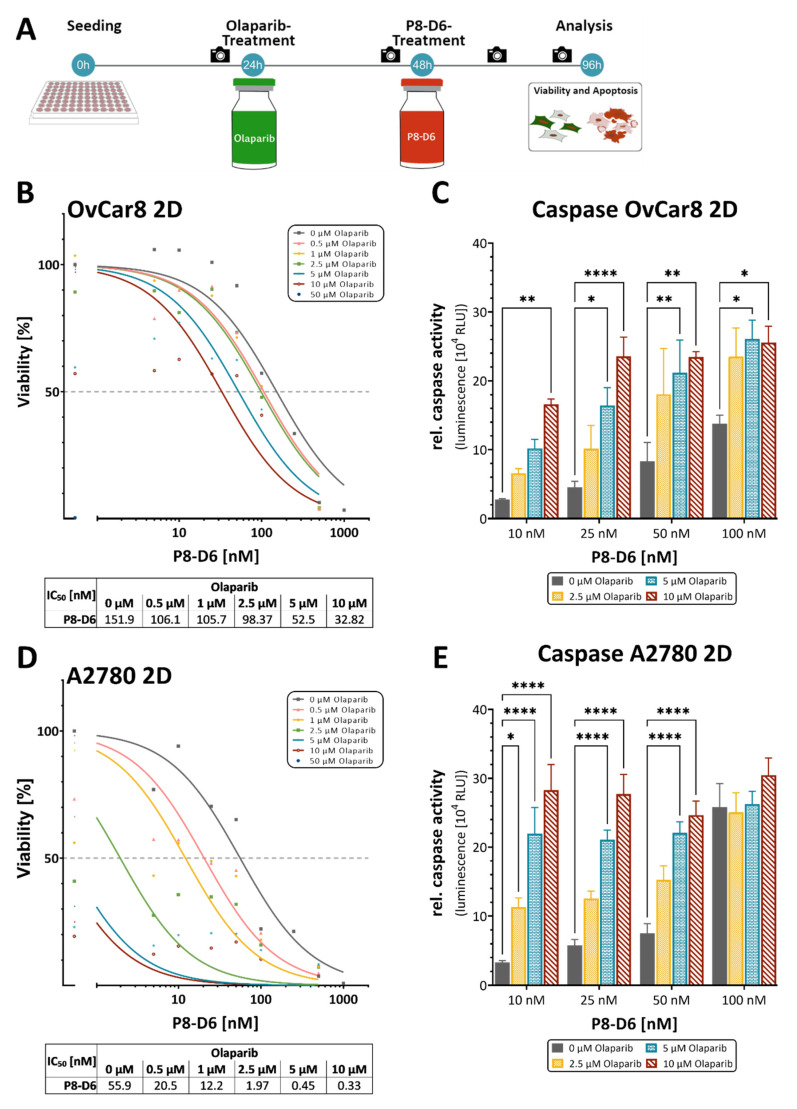
Anti-tumor responses in combined treatment of P8-D6 and olaparib in OC 2D culture. OvCar8 and A2780 cells were treated in a 2D monolayer cell culture with different concentrations of olaparib and P8-D6 in single and combination therapy. Subsequently, the viability and caspase activity were determined. (**A**) Treatment schedule. Camera icon indicates imaging timepoints. (**B**,**D**) The IC_50_ value of each olaparib concentration was calculated by using the viability data. (**C**,**E**) The apoptosis is represented as relative caspase activity. Data are means + SEM one-way ANOVA, * (*p* < 0.05), ** (*p* < 0.01), **** (*p* < 0.0001).

**Figure 3 ijms-23-10503-f003:**
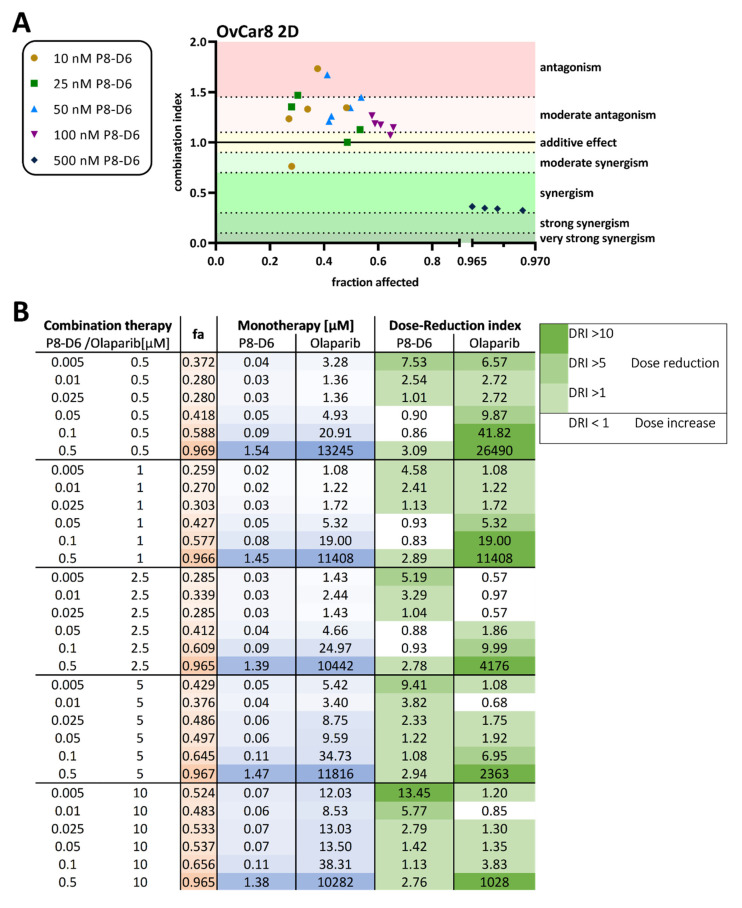
Combination index (CI) analysis of olaparib in combination with P8-D6 in OC 2D culture. (**A**) Combination index (CI) for drug combinations by olaparib and P8-D6 in OvCar8 cells. CI values computed according to CompuSyn software with viability data. Combinations were considered synergistic when CIs were below 1.0. The fraction affected (fa)-value represents the fraction of cell viability affected by therapy. (**B**) The monotherapy column defines the concentrations that are needed in monotherapy to affect a certain fraction of cells by therapy. DRI values represent the order of magnitude (fold) of dose reduction in combination setting compared with each drug alone.

**Figure 4 ijms-23-10503-f004:**
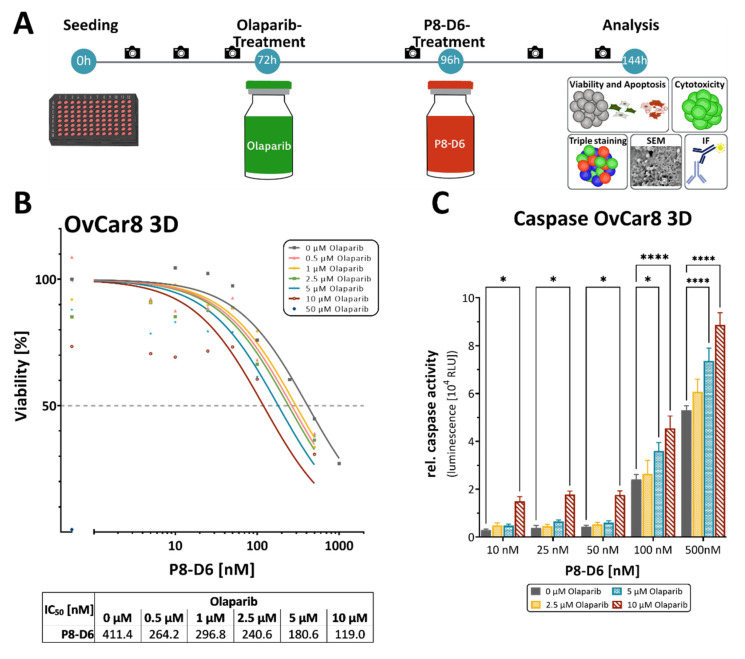
Anti-tumor responses in combined treatment of P8-D6 and olaparib in 3D culture. OvCar8 spheroids were maintained in ULA plates for 72 h and subsequently treated with olaparib and P8-D6 in single or combination treatment as illustrated in the schedule. (**A**) Treatment schedule. Camera visualizes the measurement time points by microscope. Subsequently, the viability and caspase activity were measured. (**B**) The IC_50_ value of each olaparib concentration was calculated using the viability data. (**C**) The apoptosis is represented as relative caspase activity. Data are means + SEM one-way ANOVA, * (*p* < 0.05), **** (*p* < 0.0001).

**Figure 5 ijms-23-10503-f005:**
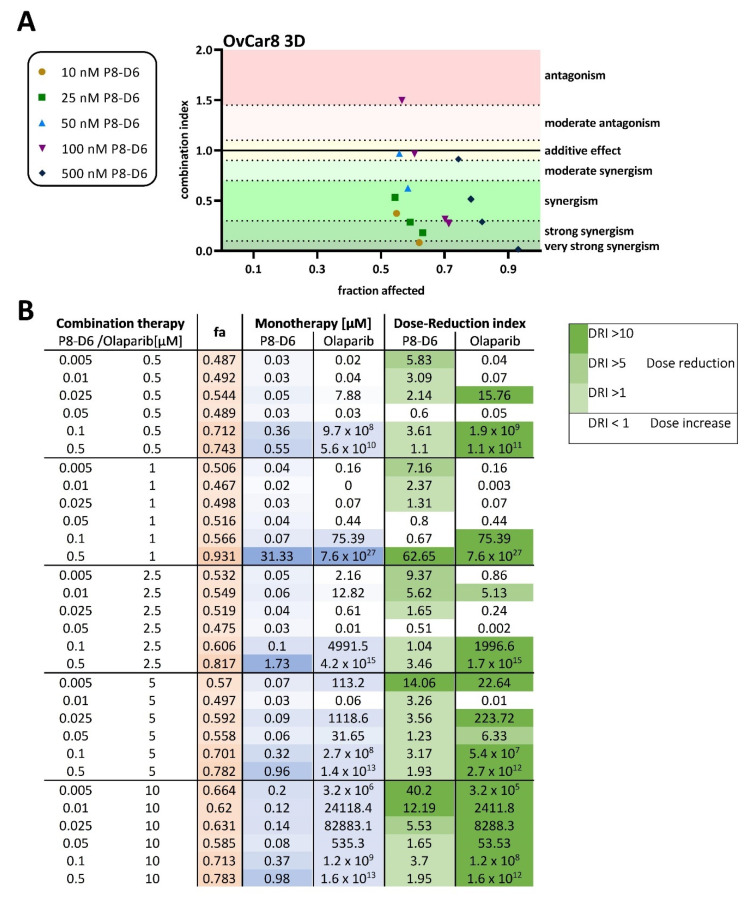
Combination index (CI) analysis of olaparib in combination with P8-D6 in OC spheroids. (**A**) Combination index (CI) for drug combinations by olaparib and P8-D6 in OvCar8 spheroids. CI values computed accordingly by CompuSyn software with viability values. Combinations were considered synergistic when CIs were below 1.0. The fraction affected (fa)-value represents the fraction of cell viability affected by therapy. (**B**) The monotherapy column defines the concentrations that are needed in monotherapy to affect a certain fraction of cells by therapy. DRI values represent the order of magnitude (fold) of dose reduction in combination setting compared with each drug alone.

**Figure 6 ijms-23-10503-f006:**
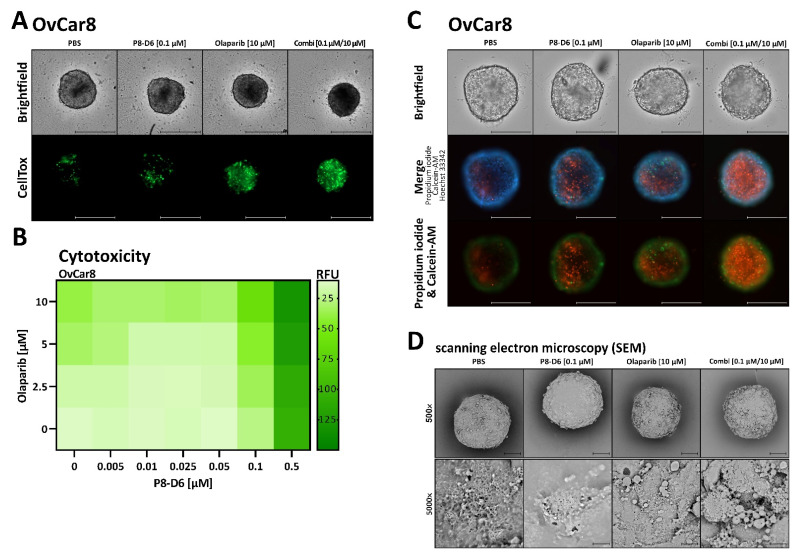
Toxicity of PARPi with dual topoisomerase inhibitor in 3D tumor spheroids of OvCar8. OvCar8 spheroids were maintained in ULA plates for 72 h and subsequently treated with olaparib and P8-D6 in single or combination treatment as described in the schedule in Figure 4A. (**A**) After treatment the cell toxicity was measured by fluorescence microscopy using CellTox™ Green (timepoint 144 h after seeding); scale bars, 500 µm. (**B**) The fluorescence signals after treatment were quantified (relative fluorescence units RFU) and shown in the heat map. (**C**) Spheroids were stained after treatment with PI (red), calcein-AM (green), Hoechst 33342 (blue) and imaged by fluorescence microscopy; scale bars, 500 µm. (**D**) Scanning electron microscopy images of spheroids, which were treated with P8-D6, olaparib, combination or PBS were taken; scale bars, 500 × 100 µm, 5000 × 10 µm.

**Figure 7 ijms-23-10503-f007:**
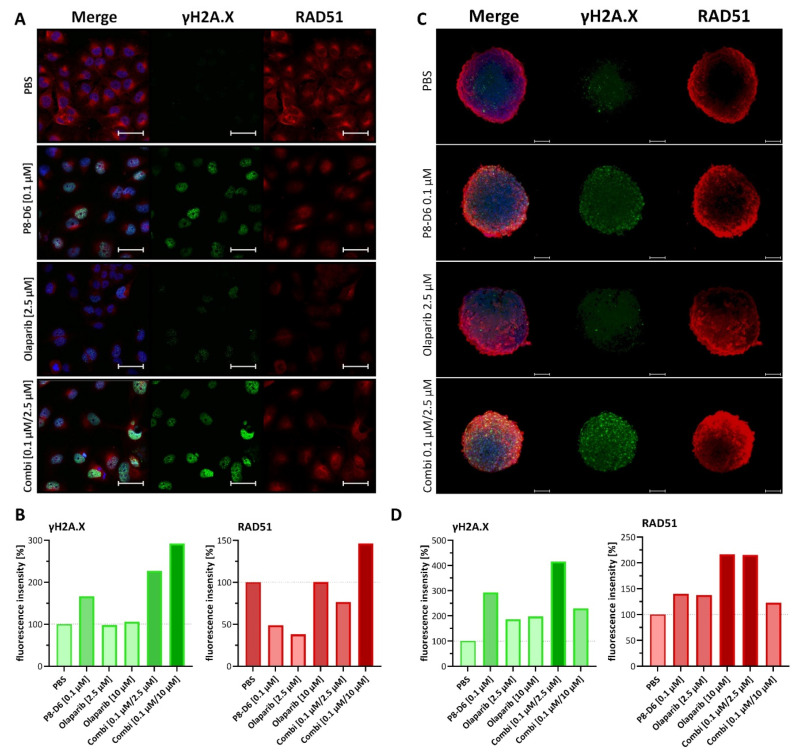
Olaparib potentiates γH2A.X and RAD51 recruitment for strand break induced by P8-D6 in OvCar8. (**A**) Immunofluorescence images of OvCar8 2D monolayer: DAPI-stained nuclei (blue), γH2A.X (green) and RAD51 (red) after treatment with olaparib and/or P8-D6; scale bars, 50 µm. (**B**) The intensity of the γH2A.X and RAD51 signal in relation to DAPI in OvCar8 2D monolayer after treatment. (**C**) Confocal fluorescence z-stack images of OvCar8 spheroids were obtained with immunostaining against γH2A.X, RAD51 and Hoechst 33342. Spheroids were treated with olaparib and/or P8-D6, permeabilized with glycerol, followed by antibody staining; scale bars, 100 µm. (**D**) γH2A.X and RAD51 fluorescence intensities of different treatments were compared between different treatments.

**Figure 8 ijms-23-10503-f008:**
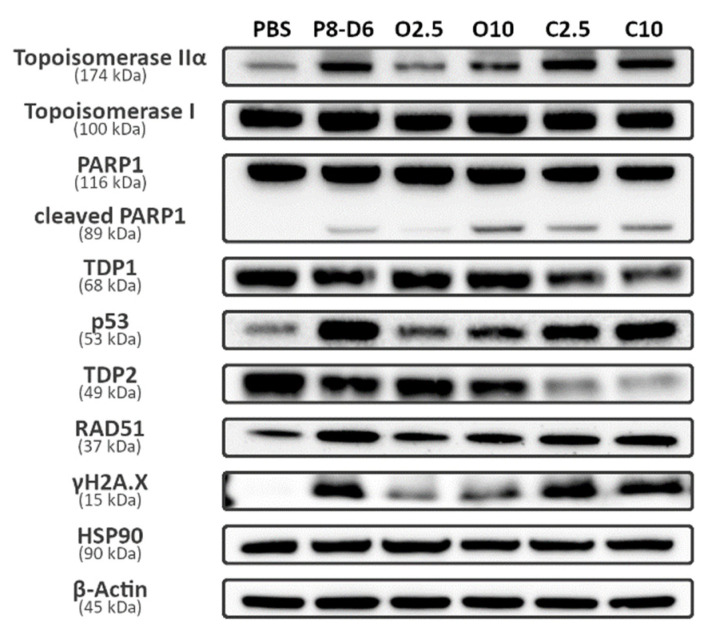
Protein expressions modulated by combinational treatment of PARP and P8-D6. Western blot analysis was performed for several cellular proteins associated with DNA damage repair systems using total OvCar8 cell lysate. HSP90 and β-Actin were used as loading control. O2.5—olaparib 2.5 µM, O10—olaparib 10 µM, C2.5—combination of olaparib 2.5 µM and P8-D6 0.1 µM, C10—combination of olaparib 10 µM and P8-D6 0.1 µM.

## Data Availability

The data presented in this study are available on request from the corresponding author.

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
