# Peer review of "Combined PARP and Dual Topoisomerase Inhibition Potentiates Genome Instability and Cell Death in Ovarian Cancer"

_ijms, 2022, doi:10.3390/ijms231810503_

Round 1

Reviewer 1 Report

The paper entitled “Combined PARP and dual topoisomerase inhibition potentiates genome instability and cell death in ovarian cancer” it is not suitable to be published in the present from.

The paper is well written, and the finding is very interesting. Authors have done an extensive and exhaustive work. Despite of this I have some concerns about the publication:

1.     About the experiment design. What did the authors decide to test only Olaparib first and then P8-D6 drug instead of administering both drugs at the same time or using P8-D6 in first instance and then Olaparib?

2.     About figure 7 panel A. It is not clear how the author justifies the effect of olaparib on 2D viability assay at 10 µM while on 3D model they show an effect at less concentration e.g. 2.5 µM.

3.     In figure 7, the confocal microscopy images should be correlate between panel A and C. Second line of panel A show P8-D6 effect whereas on the same line in panel C it has been shown the Olaparib effect. Please modify to better correlate the drug’s effects in the two different models.

4.     In figure 7 Panels B and D the RAD51 bar graph report the number of concentration value with coma instead of dot. Please correct.

5.     About figure 8 the western blot report as loading control HSP90. Usually, nuclear loading control to verify the correct nucleus lysis are TBP, histones, Laminin or PCNA. Why did the authors used another cytoplasmatic protein since they already used Actin ?

Author Response

Thank you for the opportunity to resubmit our paper after carefully answering the comments of the reviewers. First of all, we would like to thank all reviewers for their time and great input, comprehensive recommendations and helpful suggestions improving the scientific value of our paper. After a careful revision we are now confident that our paper fits perfectly into the scope of your excellent journal. We added our responses to the reviewer’s suggestions in italic (blue). All changes are highlighted in yellow in the manuscript.

  1. This is an excellent comment. Due to several pretests performed by us, in which the drugs were incubated simultaneously or with a time delay (Olaparib first, then P8-D6/ P8-D6 first, then Olaparib), we came to the decision to perform the experiments as outlined in the paper. These pre-tests showed us that an earlier administration of olaparib was more efficient and led to the best combination effects. Explanation for this may be varied and may be due to pharmacokinetic and/or pharmacodynamic processes. To make this more clear we performed changes: Line: 138
  2. In our “genome instability” experiments, the concentrations 2.5 µM olaparib and 10 µM olaparib were chosen for mono- and combination therapy in 2D and 3D culture. In 2D culture, the effect of 10 µM/0.1 µM was slightly more visible, so this concentration was selected. However, since the 10 µM/0.1 µM combination reduce membrane integrity in spheroids and makes the effects less comparable, the 2.5 µM/0.1 µM combination was now selected for 2D and 3D in the main text. The fluorescence microscopic images of the all tested concentrations were already listed for 3D in the supplement, for 2D this part is now added in the supplement. (Changes in Figure 7, Figure S4, Figure S5, Figure S6)
  3. Thank you for this feedback. The illustration has been corrected.
  4. Thank you for this feedback. The illustration has been corrected.
  5. Thanks for your comment with the loading control. We chose HSP90 as a loading control because it is commonly used as loading control and was suitable for most of our proteins due to its kDa. For analysis, total cell lysis was analysed without separation of fractions. Both controls were used only as loading controls and not as differentiation controls. Another reason for choosing two loading controls was that the individual blots were cut. We chose to detect a loading control on each cut. Also, the dual loading control served as backup in case marker bands were close to either loading control. This comment is very helpful for planning the next experiments. Future analyses will be performed with the proposed controls to determine the correct nuclear lysis together with ß-actin.

Reviewer 2 Report

I only have one question that I didn't fully understood:

·         Introduction: what do the authors mean by ‘topoisomerase poison’? Does this mean that P8-D6 is an inhibitor of the enzyme?

I suggested to accept the manuscript as I thought the article is really good. The development of new drugs to treat ovarian cancer is a key challenge in research and clinical practice as the current approaches are very toxic and, as the authors highlighted, are still not ideal in terms of efficacy, as they deal with a lot of resistances, leading to a high number of deaths because of the disease. The results are very clearly presented, I really liked the figures/graphs. 

Author Response

Thank you for the opportunity to resubmit our paper after carefully answering the comments of the reviewers. First of all, we would like to thank all reviewers for their time and great input, comprehensive recommendations and helpful suggestions improving the scientific value of our paper. After a careful revision we are now confident that our paper fits perfectly into the scope of your excellent journal. We added our responses to the reviewer’s suggestions in italic (blue). All changes are highlighted in yellow in the manuscript.

Thank you very much for this positive feedback and the appreciation of our work.

To the question what ‘topoisomerase poison’ means: Topoisomerase poisons is a specific type of topoisomerase inhibition. Topoisomerase poisons are in clinical use as anti-cancer therapy and act by covalently stabilizing the enzyme-induced DNA breaks/ covalent enzyme-cleaved DNA intermediate. In contrast, catalytic inhibitors block the enzyme before DNA scission. In line 53 the word cleaved was added.
